# Single-Point Mutations in the N Gene of SARS-CoV-2 Adversely Impact Detection by a Commercial Dual Target Diagnostic Assay

Sharon Miller,[a] Terence Lee,[a] Adam Merritt,[a] Todd Pryce,[b] Avram Levy,[a,c] David Speers[a,d]

aDepartment of Microbiology, PathWest Laboratory Medicine WA, Queen Elizabeth II Medical Centre, Nedlands, Western Australia, Australia
bDepartment of Microbiology, PathWest Laboratory Medicine WA, Fiona Stanley Hospital, Murdoch, Western Australia, Australia
cSchool of Infection and Immunity, Biomedical Sciences, University of Western Australia, Crawley, Western Australia, Australia
dSchool of Medicine, University of Western Australia, Crawley, Western Australia, Australia

**ABSTRACT** Accurate and rapid diagnostic tests are a critical component for the early diagnosis of severe acute respiratory syndrome coronavirus 2 (SARS-CoV-2) and of the overall control strategy for the current pandemic. Nucleic acid amplification tests are the gold standard for diagnosis of acute SARS-CoV-2 infection, and many real-time PCR diagnostic assays have been developed. Mutations that occur within the primer/probe binding regions of the SARS-CoV-2 genome can negatively impact the performance of diagnostic assays. Here, we report two single-point mutations in the N gene of SARS-CoV-2 associated with N gene target detection failures in the Cepheid Xpert Xpress SARS-CoV-2 assay, the first a C to T mutation at position 29197, found in five patients, and the second a C to T mutation at position 29200, found in eight patients. By sequencing the Xpert amplicons, we showed both mutations to be located within the amplified region of the Xpert N gene target. This report highlights the necessity for multiple genetic targets and the continual monitoring and evaluation of diagnostic assay performance.

**IMPORTANCE** This paper reports the identification of single-point mutations in the N gene of SARS-CoV-2 associated with a gene target failure by the Cepheid Xpert commercial system. In order to determine the mutation(s) responsible for the N gene detection failures, the genomic products from the Cepheid Xpert system were sequenced and compared to whole genomes of SARS-CoV-2 from clinical cases. This report is the first to our knowledge which characterizes the amplified PCR products of the Xpert system, confirming the mutations associated with the gene target failure. The mutations identified have previously been reported.

**KEYWORDS** SARS-CoV-2, Cepheid Xpert, mutation, diagnostic assay, N gene

Severe acute respiratory syndrome coronavirus 2 (SARS-CoV-2) originated in China in December 2019 (1) and was declared a public health emergency on 30 January 2020 (2). By 16 May 2021, SARS-CoV-2 was responsible for over 162 million confirmed cases and more than 3.3 million deaths globally (3). The publication of the SARS-CoV-2 genome on 10 January 2020 led to the rapid development of real-time PCR diagnostic tests (4) targeting the envelope (E), RNA-dependent RNA polymerase (RdRp), spike (S) protein, open reading frame (ORF), and nucleocapsid (N) genes (5). This was critical for the early diagnosis of COVID-19 to enable successful contact tracing, isolation, and quarantine measures. A number of commercial reverse transcriptase PCR (RT-PCR) tests have been produced for the diagnosis of SARS-CoV-2 infection, including the Xpert Xpress SARS-CoV-2 assay (Xpert) (Cepheid, Sunnyvale, CA, USA), which is used in many countries as a rapid laboratory test and as a point-of-care test.

Address correspondence to Sharon Miller, Sharon.miller@health.wa.gov.au.

The failure of even a single diagnostic RT-PCR in detecting SARS-CoV-2 infection can severely impair efforts to prevent and control community transmission (6). To date, three independent single-point mutations (C29200T, C29200A, and C29197T) in the N gene of the SARS-CoV-2 genome have been associated with failure to amplify the N2 gene target in the Xpert assay (7–10). A mutation in the E gene (C26340T) has also been reported, resulting in failure to amplify this target in the cobas SARS-CoV-2 assay (cobas) (Roche, Basel, Switzerland) (11).

In this report, we describe two independent single-point mutations in the N gene of SARS-CoV-2 that adversely affect the amplification of the N2 target in the Xpert assay. The first mutation, C29200T (10), was found in eight patients, and the second, C29197T (8, 9), was found in five patients. Sequencing of the Xpert amplicons confirmed the location of both mutations to be within the amplified region of the Xpert N gene target.

## RESULTS

Since April 2020, PathWest Laboratory Medicine WA has performed SARS-CoV-2 testing using the Cepheid Xpert Xpress SARS-CoV-2 assay. In late April to early May 2021, the Xpert assay failed to detect the N gene target of five samples, reporting as "presumptive positive" as the per manufacturer's instructions. In all of these cases the reverse transcriptase quantitative PCR (RT-qPCR) E gene cycle threshold ($C_T$) values were $<33$ (Table 1). The samples were confirmed positive for SARS-CoV-2 by an in-house RT-PCR using SARS-CoV-2-specific targets in the E gene (12) and the spike protein (unpublished data).

Sequencing of the positive Xpert assay cartridge showed PCR products mapped to nucleotide positions 15431 to 15530 within the ORF1ab, 26269 to 26381 within the E gene and 29164 to 29230 of the N gene, indicating the amplified regions used in the assay.

Whole-genome sequencing (WGS) of the five B.1.1.519 lineage samples from a cluster of related cases with negative N2 results on the Xpert assay identified a C29197T single nucleotide polymorphism (SNP) mutation within the probe binding region of the U.S. CDC 2019_nCoV_N2 target (Fig. 1), as previously reported by Leelawong et al. (8). The C29197T change represents a synonymous mutation of the encoded amino acid, alanine.

A retrospective analysis of our collection of SARS-CoV-2 WGS sequences revealed a set of eight B.1 lineage-related strains (from late March to mid-April 2020) with a C29200T mutation within the probe binding region similar to that previously reported by Ziegler et al. (10) (Fig. 2). Seven of the eight samples were of sufficient volume for testing using the GeneXpert system and showed similar negative results for the N2 gene target (Table 1). We were unable to test one of the clinical specimens due to insufficient volume.

**TABLE 1** Summary of $C_T$ values and sequencing data for N gene variants

| Cluster | Case no. | Date of collection | Xpert assay targets | | Lineage | N gene mutation |
| --- | --- | --- | --- | --- | --- | --- |
| | | | E gene $C_T$ | N gene $C_T$ | | |
| Cluster A | 1 | 27/03/2020 | 15.7 | 40.2 | B.1 | C29200T |
| | 2 | 04/04/2020 | 28.1 | | B.1 | C29200T |
| | 3 | 07/04/2020 | 22.6 | 42.8 | B.1 | C29200T |
| | 4 | 07/04/2020 | 23.6 | | B.1 | C29200T |
| | 5 | 08/04/2020 | 27.3 | | B.1 | C29200T |
| | 6 | 08/04/2020 | 24.4 | | B.1 | C29200T |
| | 7 | 16/04/2020 | 21.2 | | B.1 | C29200T |
| Cluster B | 1 | 26/04/2021 | 20.4 | | B.1.1.519 | C29197T |
| | 2 | 26/04/2021 | 32.8 | | B.1.1.519 | C29197T |
| | 3 | 30/04/2021 | 17.2 | | B.1.1.519 | C29197T |
| | 4 | 01/05/2021 | 16.3 | | B.1.1.519 | C29197T |
| | 5 | 01/05/2021 | 23.8 | | B.1.1.519 | C29197T |

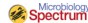

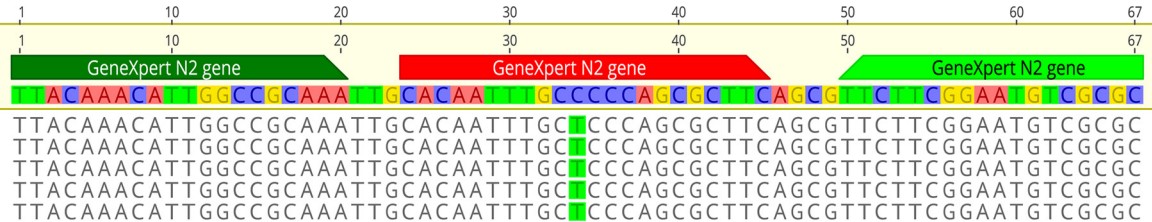

**FIG 1** Whole-genome sequencing of five samples with the C to T mutation at position 34 of the Xpert cartridge amplicon and position 29197 of the reference sequence. The top sequence shows the binding position of the U.S. CDC 2019_nCoV_N2 target. Sequences are mapped against the Wuhan-Hu-1 SARS-CoV-2 reference sequence (GenBank accession number NC_045512.2).

A GISAID search for sequences with the C29197T mutation returned 21,212 sequences, 144 of which did not have an assigned lineage. The 21,068 remaining sequences were assigned to 77 different lineages, with the B.1.1.519 lineage accounting for 91.1% of sequences. Marking the 77 lineages on a phylogenetic tree of SARS-CoV-2 indicated that the C29197T mutation was spontaneously occurring independent of evolution (Fig. 3).

## DISCUSSION

Here, we report two independent failures of the GeneXpert assay system to detect the N gene of SARS-CoV-2-positive clinical specimens, despite the E gene target being detected with relatively low $C_T$ values. The two incidences, in March 2020 and April 2021, involved two different lineages and two different mutations in the N gene target sequence. The E gene is a pan-*Sarbecovirus* target and therefore is not specific for SARS-CoV-2, prompting a "presumptive positive" report.

This study is the first to undertake sequencing of the Xpert amplicons to confirm the mutations responsible for the failure of the N gene RT-qPCR targets in the GeneXpert system; both mutations identified in this study have previously been reported (8–10). As the primers and probe sequences used by the Xpert assay are not publicly available, the amplified products of the Xpert assay were sequenced to identify the targets. The targets were then used to infer the mutation(s) most likely responsible for the failure of the N gene detection. Sequencing successfully identified the Xpert assay targets pertaining to the N gene, E gene, and ORF1a region of the SARS-CoV-2 genome. Based on the amplicon sequences, the N2 gene target region used by the Xpert was found to be consistent with the 2019_nCoV_N2 probe sequences published by the CDC, as indicated by Ziegler et al. (10). Alignment of the N gene target sequence with the five sample sequences identified a C to T mutation at location 29197, indicating that this mutation in the probe binding region is likely responsible for the failed detection of the N gene by the Xpert assay, similarly reported by Leelawong et al. (8).

The N gene target sequence was further aligned with all SARS-CoV-2 sequences in the PathWest database, and an additional eight sequences were found to contain a C to T mutation at position 29200, within the CDC probe binding region. These samples

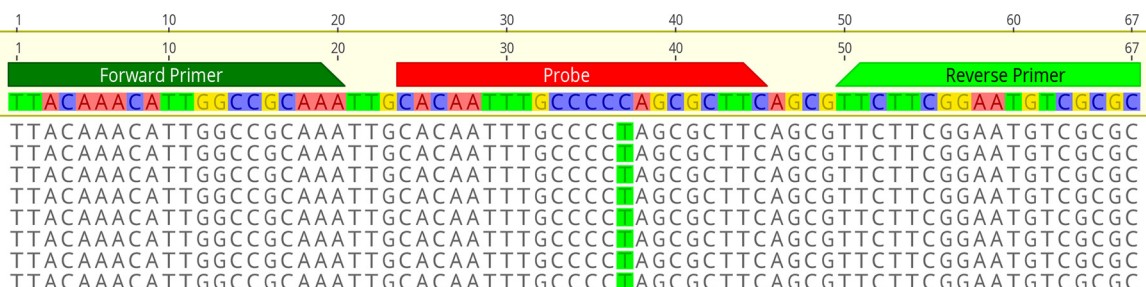

**FIG 2** Whole-genome sequencing of five samples with the C to T mutation at position 29200. The top sequence shows the binding position of the U.S. CDC 2019_nCoV_N2 target. Sequences are mapped against the Wuhan-Hu-1 SARS-CoV-2 reference sequence (GenBank accession: NC_045512.2).

Tree scale: 1

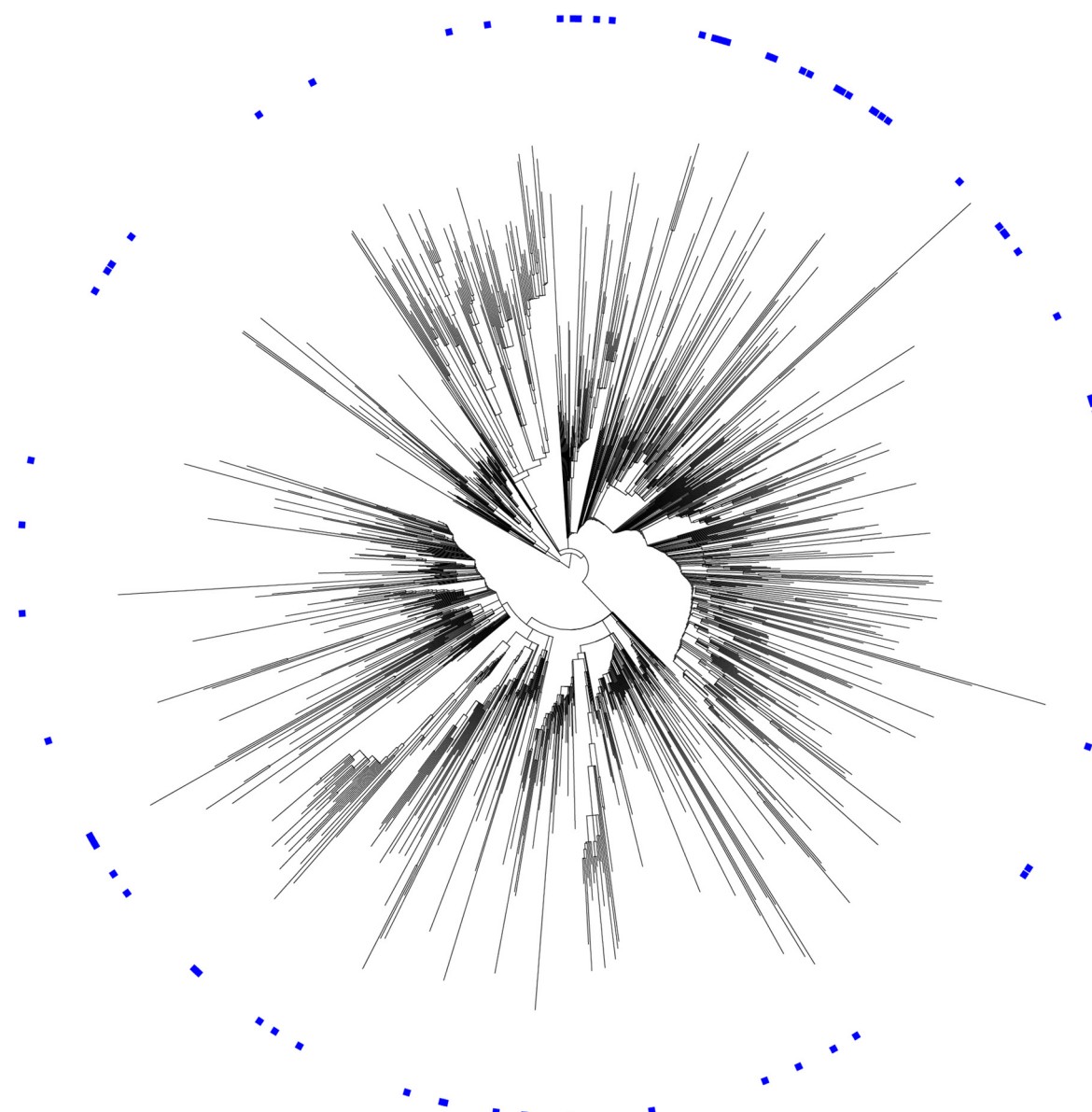

**FIG 3** Phylogenetic tree of SARS-CoV-2 showing the distribution of sequences containing the C29197T mutation (in blue).

isolated in March and April of 2020 were also from a local cluster of a single lineage but had not been previously tested on the GeneXpert assay system. It was hypothesized that the C29200T mutation in these strains would also negatively affect the Xpert detection of the N gene target. Testing of the clinical specimens on the Xpert assay resulted in an N gene target failure despite low E gene $C_T$ values. It is important to note that in two of the samples containing the C29200T mutation, with two of the lowest E gene $C_T$ values, the N gene was detected with very high $C_T$ values, >40. This is indicative of poor amplification or probe efficiency for these two samples, likely a result of the C29200T mutation.

Our results show that the N2 gene target region used by the Xpert is consistent with the CDC 2019_nCoV_N2 probe sequence. This is significant, as other commercial or in-house diagnostic assays designed using this sequence may encounter similar issues. The C29197T and C29200T mutations, located within the CDC probe sequence,

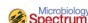

are likely responsible for the failed detection of the N gene target in the Xpert assay and have the potential to negatively impact detection in other assays which also use this probe sequence.

At present, three mutations have been identified in the N gene that likely result in N gene target detection failures in the Xpert assay; C29197T (8, 9), C29200T (10), and C29200A (7). All three mutations affect the poly C region within the probe binding region of the U.S. CDC 2019_nCoV_N2 target, indicating that mutations in this region may be associated with poor probe performance. The C29200T mutation is widespread, accounting for more than 0.2% of all sequences uploaded to the GISAID EpiCoV database as of September 2020 (10). In this study, the five isolates containing the C29197T mutation were from a cluster of related cases. Analysis of these sequences using the GISAID database showed them to be closely related to sequences isolated in Colorado, USA. Rhoads et al. (9) and Leelawong et al. (8) also identified the C29197T mutation in cases from Ohio and New York, respectively. The C29197T mutation accounts for approximately 0.01% of all sequences uploaded to the GISAID EpiCoV database as of June 2021. A study by Rhoads et al. (9) also found the C29197T mutation to be present in less than 1% of sequences in the NCBI and GISAID databases. Further analysis revealed that the C29197T mutation is present in 77 SARS-CoV-2 lineages, indicating that it is likely the result of spontaneous mutation independent of evolution. Interestingly, the most frequent mutations to occur in the SARS-CoV-2 genome are C to T; this mutational bias is linked to the role of cell-derived apolipoprotein B mRNA editing-enzyme, which induces C to U and G to A mutations (13). This can account for the spontaneous appearance of SNPs, and it is therefore not surprising that the C29197T and C29200T mutations are widespread in the GISAID EpiCoV database, occurring in different lineages and in separate transmission chains.

A study by Wang et al. (6) found mutations in all primer/probe binding regions of published SARS-CoV-2 diagnostic targets. The majority of diagnostic assays are designed to detect one or more of three regions in the genome, the RdRp gene in ORF1ab, the E gene, and the N gene. All three regions have accumulated significant mutations since the initial isolation of SARS-CoV-2, with the N gene having the highest number of mutations per residue (14). Interestingly, analysis of SNPs in the E gene found the C26340T mutation to be the most frequent (14). This mutation has been associated with failure of the cobas SARS-CoV-2 E gene RT-qPCR (11).

Mutations put at risk detection assays that rely on a solitary target in the viral genome. It is also prudent that diagnostic design account for these potential changes. It has become widely known that mutations within the SARS-CoV-2 genome can negatively impact the performance of diagnostic assays, and recently, the FDA released recommendations to all developers on design considerations and routine monitoring of viral mutations (15). This has become increasingly more important with the development and rollout of the 4-plex Xpert assay. Diagnostic laboratories will no longer be able to detect and report on single target failures, as the assay provides a single positive result from the amplification of either the N gene, the E gene, or both.

The results from this study highlight the necessity of multiple genetic targets and supports the need for continued surveillance of SARS-CoV-2 evolution and primer/probe review for diagnostic assay performance. False-negative tests due to viral variants that can no longer be detected by routine diagnostic assays are a real possibility with the potential for serious consequences in the management of the pandemic.

## MATERIALS AND METHODS

**Xpert testing.** Briefly, 300 $\mu$L of a throat and bilateral nose swab elution in viral transport medium was loaded into an Xpert cartridge, and the test was conducted using the GeneXpert Infinity system. A positive result was reported if the N2 target alone was detected or if both N2 and E targets were detected. A presumptive positive result was reported if only the E target was detected.

**Retrieval of amplicon from the Xpert cartridge.** In a separate facility from where routine Xpert testing was performed, used SARS-CoV-2 RNA-positive and -negative Xpert cartridges were placed in a class II biosafety cabinet (BSC). The reaction reading chamber was carefully removed with sterile scissors. Using a fine 10 $\mu$L pipette tip, the contents of the reaction chamber were aspirated, and approximately

50 µL of reaction product was transferred to a microcentrifuge tube. Then, 10 µL of reaction product was combined with 10 µL of PCR-grade water in a 96-well plate, and the reaction products were analyzed with Perkin Elmer LabChip GXII using a Hisense DNA kit in a third facility.

**Sequencing of Xpert cartridge amplicons.** The TruSeq chromatin immunoprecipitation (ChIP) library preparation kit was chosen due to the size and concentration of the input DNA. The TruSeq ChIP sample preparation protocol used a DNA input concentration of 5 to 10 ng in a 50 µL volume, amplification products were normalized to 200 pg/µL in a volume of 50 µL in a 96-well plate, and then end-repair was performed to convert overhangs into blunt ends and phosphorylate the 5′ terminal. Incubation was for 30 min at 30°C in a Kyratec SuperCycler thermocycler, which was used for all subsequent incubation and amplification steps. A bead cleanup was performed using AMPure XP beads (85% ethanol concentration). The 3′ ends were adenylated to prevent ligation to one another, with incubation at 37°C for 30 min followed by 70°C for 5 min. Next, adapter ligation was performed with TruSeq single indexes for subsequent multiplex sequencing. The ligation reaction incubation was 30°C for 10 min. Two sequential bead cleanups followed with a volume of AMPure XP beads to sample ratio of 2:1 and 1.4:1, respectively. In both cleanups the volume of AMPure XP beads was increased to preserve the smaller DNA fragments. Ligation products were purified by gel electrophoresis on a 3% agarose gel, and band excision was performed using the Qiagen MinElute gel extraction kit following the manufacturer's instructions. Due to the adapter ligation step, the expected products were approximately 100 to 120 bp larger than the original DNA fragments. To avoid primer-dimers and adapter-dimers (estimated to be 120 to 140 bp in size), only products between 150 and 250 bp were excised. For each sample, two gel slices were excised spanning approximately 150 to 200 bp and 200 to 250 bp, and excision was performed using a 50 bp and a 100 bp ladder for size estimation. The two gel fragments for each sample were processed separately with the Qiagen MinElute gel extraction kit and the subsequent enrichment step. The enrichment step was done to selectively amplify the adapter-bound amplicons to increase the amount of DNA in the library prior to sequencing. The PCR parameters were 98°C for 30 sec, 18 cycles of 98°C for 10 sec, 60°C for 30 sec, and 72°C for 30 sec, followed by 72°C for 5 min and hold at 4°C. During the final AMPure XP bead cleanup, the PCR products for each sample were combined; the AMPure XP bead to sample ratio was 1:1.

Following library preparation, the libraries were quantified using the Qubit 1× double-stranded DNA (dsDNA) high-sensitivity (HS) assay kit (Invitrogen) and the Agilent TapeStation 4200 on D5000 ScreenTapes following the manufacturer's instructions. Libraries were normalized to 2 nM and pooled. Pooled libraries were diluted and denatured to a final concentration of 7 pM and sequenced using the Illumina MiSeq system and micro V2 cartridge.

**RNA extraction and whole-genome sequencing of SARS-CoV-2 from sample.** RNA was extracted from clinical samples using the OMEGA Mag-Bind RNA Xpress kit following the manufacturer's instructions. Reverse transcription was preformed using SuperScript IV VILO MasterMix (Thermo Fisher Scientific). Briefly, 5 µL of viral RNA was added to 5 µL of MasterMix and 15 µL of $H_2O$ with incubation at 25°C for 10 min, 50°C for 25 min, and 85°C for 5 min. The viral cDNA was used to generate 1,000 bp SARS-CoV-2 tiled amplicons, as described previously (16). Amplicons were pooled, purified, and quantified. Nextera XT libraries were prepared and sequenced on an Illumina iSeq sequencer.

**Bioinformatics analysis.** To determine the Xpert assay target regions, raw sequencing reads obtained from sequencing the positive Xpert assay cartridge were mapped using Burrows-Wheeler Aligner (BWA) v0.7.17 (17) to the Wuhan-Hu-1 SARS-CoV-2 reference sequence (GenBank accession number NC_045512.2). The mapped regions were visualized and reviewed in Geneious Prime v2021.0.1.

To determine the lineage of samples, WGS reads were mapped to the SARS-CoV-2 reference sequence, sorted using SAMtools v1.9 (18), and reviewed in Geneious Prime v2021.0.1. A consensus sequence for each sample was generated using a 95% threshold, and lineages were assigned to each sequence using Pangolin v3.1.11 (https://github.com/cov-lineages/pangolin).

GISAID sequences (19) were retrieved on 16 June 2021 and queried for the subsequence TTTGCTCCCAGCGC consisting of the identified mutation using a custom script. The results were aligned to the reference sequence and reviewed in Geneious Prime. Sequences consisting of the mutation were assigned a lineage as described previously.

All nucleotide positions were referenced to the Wuhan-Hu-1 SARS-CoV-2 reference sequence (GenBank accession number NC_045512.2).

**Data availability.** The whole-genome sequences are available on GISAID under accession numbers EPI_ISL_1816918, EPI_ISL_1828698, EPI_ISL_1828700, EPI_ISL_1914666, EPI_ISL_1914667, EPI_ISL_470858, EPI_ISL_470859, EPI_ISL_470860, EPI_ISL_470861, EPI_ISL_470862, EPI_ISL_512756, EPI_ISL_512757, and EPI_ISL_512758.

## ACKNOWLEDGMENT

We thank and acknowledge Marina Donskoi (field applications scientist, Illumina) for her expertise and assistance in developing the methodology for this paper.

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
