## [Reviewer comments · Microbiology Spectrum]

Microbiology Spectrum

Single-point mutations in the N gene of SARS-CoV-2 adversely impact detection by a commercial dual target diagnostic assay

Sharon Miller, Terence Lee, Adam Merritt, Todd Pryce, Avram Levy, and David Speers

Corresponding Author(s): Sharon Miller, PathWest Laboratory Medicine

Review Timeline:

Submission Date:	September 5, 2021
Editorial Decision:	October 4, 2021
Revision Received:	October 12, 2021
Editorial Decision:	October 18, 2021
Revision Received:	October 21, 2021
Accepted:	October 25, 2021

Editor: Yun Young Go

Reviewer(s): The reviewers have opted to remain anonymous.

Transaction Report:

DOI: <https://doi.org/10.1128/Spectrum.01494-21>

October 4, 2021

Dr. Sharon Anne Miller
PathWest Laboratory Medicine
Microbiology
Queen Elizabeth II Medical Centre
Nedlands
Perth, WA 6009
Australia

Re: Spectrum01494-21 (Single-point mutations in the N gene of SARS-CoV-2 adversely impact detection by a commercial dual target diagnostic assay)

Dear Dr. Sharon Anne Miller:

Thank you for submitting your manuscript to Microbiology Spectrum. Your manuscript has been reviewed by two experts in the field. As their comments indicate, the manuscript needs modifications before it could be considered acceptable for publication. When submitting the revised version of your paper, please provide (1) point-by-point responses to the issues raised by the reviewers as file type "Response to Reviewers," not in your cover letter, and (2) a PDF file that indicates the changes from the original submission (by highlighting or underlining the changes) as file type "Marked Up Manuscript - For Review Only". Please use this link to submit your revised manuscript - we strongly recommend that you submit your paper within the next 60 days or reach out to me. Detailed information on submitting your revised paper are below.

Link Not Available

Sincerely,

Yun Young Go

Journals Department
Reviewer comments:

Reviewer #1 (Comments for the Author):

This manuscript is a description of mutations in the N2 region of SARS-CoV-2 that impact one of the targets in the Cepheid assay. These mutations have been described previously, however the novelty in this work is the geographic location of the discovery (which could be more specifically noted in the manuscript, see below) and the analysis that was performed. This group sequenced the products from the GenXpert cartridge and did genomic analyses that were not previously described.

Two comments/suggestions:

1. Note the location of the patients from whom these specimens were derived and compare to previously publications. The emergence of these mutations on another continent adds to the genomics analyses and the hypothesis that these mutations are independently emerging.

2. Please check the mutation notation in line 167 of the manuscript. I think this should read "C29297T" not "C28932T".

Reviewer #2 (Comments for the Author):

The manuscript entitled "Single-point mutations in the N gene of SARS-CoV-2 adversely impact detection by a commercial dual target diagnostic assay" by Miller and colleagues described mutations in the N gene of SARS-CoV-2 that may affect the detection of the virus by the Cepheid Xpert commercial system.

This paper reports the identification of single-point mutations in the N gene of SARS-CoV-2 associated with a gene target failure by the Cepheid Xpert commercial system. To determine the mutation(s) responsible for the N gene detection failures, the genomic products from the Cepheid Xpert system were sequenced and compared to whole genomes of SARS-CoV-2 from clinical cases. However, the manuscript has some concerns.

Comment 1: Line 43 and Line 197: "This report is the first to our knowledge which characterises the amplified PCR products of the Xpert system and identifies the mutations associated with the gene target failure," and "This study is the first to undertake sequencing of the Xpert amplicons to confirm the mutations responsible for the failure of the N gene RT-qPCR targets in the GeneXpert system." As previous studies already reported these mutations, these statement can be revised.

Comment 2: Line 66: Please put a comma after "In this report"

Comment 3: Line 146: "In late April to early May 2021, the Xpert assay failed to detect the N gene target of five samples, reporting as "presumptive positive" as per manufacturer's instructions. In all of these cases the RT-qPCR E gene cycle threshold (CT) values were < 33 (Table 1)." -Are these samples being tested by any other commercial or in-house assays?

Comment 4: It would be helpful for readers if the sequences were deposited in GISAID or GenBank and mentioned it in data availability section.

Comment 5: Laboratory researchers would be interested to know if the described mutation could possibly affect (in silico experiment) any of the commonly used commercial/in-house diagnostic assays?

Staff Comments:

Preparing Revision Guidelines

Please return the manuscript within 60 days; if you cannot complete the modification within this time period, please contact me. If you do not wish to modify the manuscript and prefer to submit it to another journal, please notify me of your decision immediately so that the manuscript may be formally withdrawn from consideration by Microbiology Spectrum.

The manuscript entitled "Single-point mutations in the N gene of SARS-CoV-2 adversely impact detection by a commercial dual target diagnostic assay" by Miller and colleagues described mutations in the N gene of SARS-CoV-2 that may affect the detection of the virus by the Cepheid Xpert commercial system.

This paper reports the identification of single-point mutations in the N gene of SARS-CoV-2 associated with a gene target failure by the Cepheid Xpert commercial system. To determine the mutation(s) responsible for the N gene detection failures, the genomic products from the Cepheid Xpert system were sequenced and compared to whole genomes of SARS-CoV-2 from clinical cases. However, the manuscript has some concerns.

Comment 1: Line 43 and Line 197: "This report is the first to our knowledge which characterises the amplified PCR products of the Xpert system and identifies the mutations associated with the gene target failure," and "This study is the first to undertake sequencing of the Xpert amplicons to confirm the mutations responsible for the failure of the N gene RT-qPCR targets in the GeneXpert system."- As previous studies already reported these mutations, these statement can be revised.

Comment 2: Line 66: Please put a comma after "In this report"

Comment 3: Line 146: "In late April to early May 2021, the Xpert assay failed to detect the N gene target of five samples, reporting as "presumptive positive" as per manufacturer's instructions. In all of these cases the RT-qPCR E gene cycle threshold (CT) values were < 33 (Table 1)." -Are these samples being tested by any other commercial or in-house assays?

Comment 4: It would be helpful for readers if the sequences were deposited in GISAID or GenBank and mentioned it in data availability section.

Comment 5: Laboratory researchers would be interested to know if the described mutation could possibly affect (in silico experiment) any of the commonly used commercial/in-house diagnostic assays?

RESPONSE TO REVIEWERS

Reviewer #1 (Comments for the Author):

This manuscript is a description of mutations in the N2 region of SARS-CoV-2 that impact one of the targets in the Cepheid assay. These mutations have been described previously, however the novelty in this work is the geographic location of the discovery (which could be more specifically noted in the manuscript, see below) and the analysis that was performed. This group sequenced the products from the GenXpert cartridge and did genomic analyses that were not previously described.

Two comments/suggestions:

1. Note the location of the patients from whom these specimens were derived and compare to previously publications. The emergence of these mutations on another continent adds to the genomics analyses and the hypothesis that these mutations are independently emerging.

In this study, the five isolates containing the C29197T mutation were from a cluster of related cases. Analysis of these sequences using the GISAID database showed them to be closely related to sequences isolated in Colorado, USA. Rhoads et al. (9) and Leelawong et al. (8) also identified the C29197T mutation in cases from Ohio and New York, respectively.

2. Please check the mutation notation in line 167 of the manuscript. I think this should read "C29297T" not "C28932T".

Correction made

Reviewer #2 (Comments for the Author)

The manuscript entitled "Single-point mutations in the N gene of SARS-CoV-2 adversely impact detection by a commercial dual target diagnostic assay" by Miller and colleagues described mutations in the N gene of SARS-CoV-2 that may affect the detection of the virus by the Cepheid Xpert commercial system.

This paper reports the identification of single-point mutations in the N gene of SARS-CoV-2 associated with a gene target failure by the Cepheid Xpert commercial system. To determine the mutation(s) responsible for the N gene detection failures, the genomic products from the Cepheid Xpert system were sequenced and compared to whole genomes of SARS-CoV-2 from clinical cases. However, the manuscript has some concerns.

Comment 1: Line 43 and Line 197: "This report is the first to our knowledge which characterises the amplified PCR products of the Xpert system and identifies the mutations associated with the gene target failure," and "This study is the first to undertake sequencing of the Xpert amplicons to confirm the mutations responsible for the failure of the N gene RT-qPCR targets in the GeneXpert system."- As previous studies already reported these mutations, these statement can be revised.

Wording has been revised.

Line 44 "confirming the mutations associated with the gene target failure. The mutations identified have previously been reported."

Line 197 “This study is the first to undertake sequencing of the Xpert amplicons to confirm the mutations responsible for the failure of the N gene RT-qPCR targets in the GeneXpert system, both mutations identified in this study have previously been reported (8-10).”

Comment 2: Line 66: Please put a comma after “In this report”

Correction made

Comment 3: Line 146: “In late April to early May 2021, the Xpert assay failed to detect the N gene target of five samples, reporting as “presumptive positive” as per manufacturer’s instructions. In all of these cases the RT-qPCR E gene cycle threshold (CT) values were < 33 (Table 1).” -Are these samples being tested by any other commercial or in-house assays?

Yes, these samples were confirmed with an in-house RT-PCR system.

Comment 4: It would be helpful for readers if the sequences were deposited in GISAID or GenBank and mentioned it in data availability section.

A data availability section consisting of the GISAID Accession numbers has been added.

Comment 5: Laboratory researchers would be interested to know if the described mutation could possibly affect (in silico experiment) any of the commonly used commercial/in-house diagnostic assays?

Yes, the described mutation could possibly affect the ability of the gene Xpert kit in detecting the N gene target of SARS-CoV-2.

October 18, 2021

Dr. Sharon Anne Miller
PathWest Laboratory Medicine
Microbiology
Queen Elizabeth II Medical Centre
Nedlands
Perth, WA 6009
Australia

Re: Spectrum01494-21R1 (Single-point mutations in the N gene of SARS-CoV-2 adversely impact detection by a commercial dual target diagnostic assay)

Dear Dr. Sharon Anne Miller:

In the revised version, the authors have made an effort to improve the manuscript following the reviewers' comments. However, comments 3 and 5 from Reviewer #2 still need additional clarification.

Specific comments:

1. Page 7, lines 152-153. Please provide specifics of the in-house RT-PCR performed in the study (e.g., target gene) and the reference material.
2. The possible impact of the described mutation in commonly used commercial or in-house diagnostic assays (other than Xpert assay), as suggested by reviewer #2 should be discussed thoroughly.

Thank you for submitting your manuscript to Microbiology Spectrum. When submitting the revised version of your paper, please provide (1) point-by-point responses to the issues raised by the reviewers as file type "Response to Reviewers," not in your cover letter, and (2) a PDF file that indicates the changes from the original submission (by highlighting or underlining the changes) as file type "Marked Up Manuscript - For Review Only". Please use this link to submit your revised manuscript - we strongly recommend that you submit your paper within the next 60 days or reach out to me. Detailed information on submitting your revised paper are below.

Link Not Available

Sincerely,

Yun Young Go

Journals Department
Staff Comments:

Preparing Revision Guidelines

Please return the manuscript within 60 days; if you cannot complete the modification within this time period, please contact me. If you do not wish to modify the manuscript and prefer to submit it to another journal, please notify me of your decision immediately so that the manuscript may be formally withdrawn from consideration by Microbiology Spectrum.

RESPONSE TO REVIEWERS

Reviewer #1 (Comments for the Author):

This manuscript is a description of mutations in the N2 region of SARS-CoV-2 that impact one of the targets in the Cepheid assay. These mutations have been described previously, however the novelty in this work is the geographic location of the discovery (which could be more specifically noted in the manuscript, see below) and the analysis that was performed. This group sequenced the products from the GenXpert cartridge and did genomic analyses that were not previously described.

Two comments/suggestions:

1. Note the location of the patients from whom these specimens were derived and compare to previously publications. The emergence of these mutations on another continent adds to the genomics analyses and the hypothesis that these mutations are independently emerging.

In this study, the five isolates containing the C29197T mutation were from a cluster of related cases. Analysis of these sequences using the GISAID database showed them to be closely related to sequences isolated in Colorado, USA. Rhoads et al. (9) and Leelawong et al. (8) also identified the C29197T mutation in cases from Ohio and New York, respectively.

2. Please check the mutation notation in line 167 of the manuscript. I think this should read "C29297T" not "C28932T".

Correction made

Reviewer #2 (Comments for the Author)

The manuscript entitled "Single-point mutations in the N gene of SARS-CoV-2 adversely impact detection by a commercial dual target diagnostic assay" by Miller and colleagues described mutations in the N gene of SARS-CoV-2 that may affect the detection of the virus by the Cepheid Xpert commercial system.

This paper reports the identification of single-point mutations in the N gene of SARS-CoV-2 associated with a gene target failure by the Cepheid Xpert commercial system. To determine the mutation(s) responsible for the N gene detection failures, the genomic products from the Cepheid Xpert system were sequenced and compared to whole genomes of SARS-CoV-2 from clinical cases. However, the manuscript has some concerns.

Comment 1: Line 43 and Line 197: "This report is the first to our knowledge which characterises the amplified PCR products of the Xpert system and identifies the mutations associated with the gene target failure," and "This study is the first to undertake sequencing of the Xpert amplicons to confirm the mutations responsible for the failure of the N gene RT-qPCR targets in the GeneXpert system."- As previous studies already reported these mutations, these statement can be revised.

Wording has been revised.

Line 44 "confirming the mutations associated with the gene target failure. The mutations identified have previously been reported."

Line 197 “This study is the first to undertake sequencing of the Xpert amplicons to confirm the mutations responsible for the failure of the N gene RT-qPCR targets in the GeneXpert system, both mutations identified in this study have previously been reported (8-10).”

Comment 2: Line 66: Please put a comma after “In this report”

Correction made

Comment 3: Line 146: “In late April to early May 2021, the Xpert assay failed to detect the N gene target of five samples, reporting as “presumptive positive” as per manufacturer’s instructions. In all of these cases the RT-qPCR E gene cycle threshold (CT) values were < 33 (Table 1).” -Are these samples being tested by any other commercial or in-house assays?

The samples were confirmed positive for SARS-CoV-2 by an in-house RT-PCR using SARS-CoV-2 specific targets in the E gene (16) and the spike protein (unpublished data).

Comment 4: It would be helpful for readers if the sequences were deposited in GISAID or GenBank and mentioned it in data availability section.

A data availability section consisting of the GISAID Accession numbers has been added.

Comment 5: Laboratory researchers would be interested to know if the described mutation could possibly affect (in silico experiment) any of the commonly used commercial/in-house diagnostic assays?

The following paragraph has been added to the discussion

Our results show that the N2 gene target region used by the Xpert is consistent with the CDC 2019_nCoV_N2 probe sequence. This is significant as other commercial or in-house diagnostic assays designed using this sequence may encounter similar issues. The C29197T and C29200T mutations, located within the CDC probe sequence, are likely responsible for the failed detection of the N gene target in the Xpert assay and have the potential to negatively impact detection in other assays which also use this probe sequence.

October 25, 2021

Dr. Sharon Anne Miller
PathWest Laboratory Medicine
Microbiology
Queen Elizabeth II Medical Centre
Nedlands
Perth, WA 6009
Australia

Re: Spectrum01494-21R2 (Single-point mutations in the N gene of SARS-CoV-2 adversely impact detection by a commercial dual target diagnostic assay)

Dear Dr. Sharon Anne Miller:

Your manuscript has been accepted, and I am forwarding it to the ASM Journals Department for publication. You will be notified when your proofs are ready to be viewed.

Sincerely,

Yun Young Go
Editor, Microbiology Spectrum
